# Visual Instruction Tuning with 500x Fewer Parameters through Modality Linear Representation-Steering

## Abstract

Multimodal Large Language Models (MLLMs) have significantly advanced visual tasks by integrating visual representations into large language models (LLMs). The textual modality, inherited from LLMs, equips MLLMs with abilities like instruction following and in-context learning. In contrast, the visual modality enhances performance in downstream tasks by leveraging rich semantic content, spatial information, and grounding capabilities. These intrinsic modalities work synergistically across various visual tasks. Our research initially reveals a persistent imbalance between these modalities, with text often dominating output generation during visual instruction tuning. This imbalance occurs when using both full fine-tuning and parameter-efficient fine-tuning (PEFT) methods. We then found that re-balancing these modalities can significantly reduce the number of trainable parameters required, inspiring a direction for further optimizing visual instruction tuning. Hence, in this paper, we introduce Modality Linear Representation-Steering (MoReS) to achieve the goal. MoReS effectively re-balances the intrinsic modalities throughout the model, where the key idea is to steer visual representations through linear transformations in the visual subspace across each model layer. To validate our solution, we composed LLaVA Steering, a suite of models integrated with the proposed MoReS method. Evaluation results show that the composed LLaVA Steering models require, on average, 500 times fewer trainable parameters than LoRA needs while still achieving comparable performance across three visual benchmarks and eight visual question-answering tasks. Last, we present the LLaVA Steering Factory, an in-house developed platform that enables researchers to quickly customize various MLLMs with component-based architecture for seamlessly integrating state-of-the-art models, and evaluate their intrinsic modality imbalance. This open-source project enriches the research community to gain a deeper understanding of MLLMs.

## 1 Introduction

Recent advancements in Multimodal Large Language Models (MLLMs) (Liu et al., 2024b; Xue et al., 2024; Zhou et al., 2024a; Chen et al., 2023) have demonstrated impressive capabilities across a variety of visual downstream tasks. These models integrate visual representations from pretrained vision encoders via various connectors (Liu et al., 2024a; Li et al., 2023a; Alayrac et al., 2022) into LLMs, leveraging the latter's sophisticated reasoning abilities (Zhang et al., 2024; Abdin et al., 2024; Zheng et al., 2023a).

To better integrate visual representations into LLMs, the most popular MLLMs adopt a two-stage training paradigm: pretraining followed by visual instruction tuning. In the pretraining stage, a connector is employed to project visual representations into the textual representation space. We define these two modalities—text and vision—as intrinsic to MLLMs, each carrying rich semantic information that serves as the foundation for further visual instruction tuning on downstream tasks

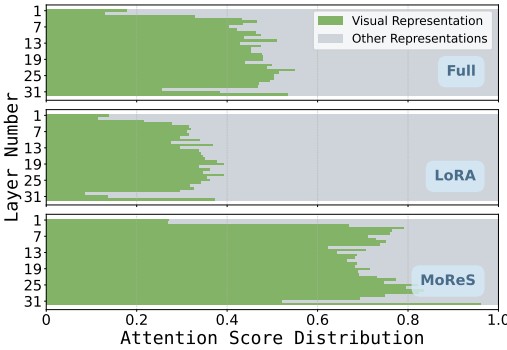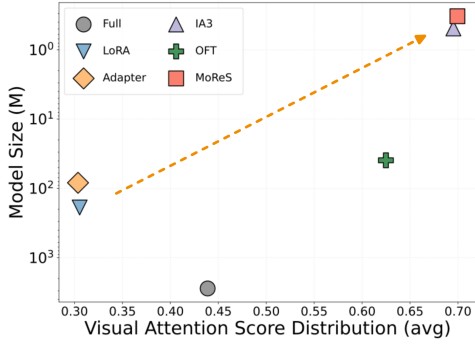

Figure 1: **Left:** Attention score distributions across layers for three MLLM fine-tuning methods (Full, LoRA, and MoReS), sampled from 100 instances each. Green represents visual representations, while grey indicates other (primarily textual) representations. Full fine-tuning and LoRA show strong reliance on textual representations across most layers. In contrast, the proposed MoReS method demonstrates significantly improved visual representation utilization, particularly in the middle and lower layers, addressing the intrinsic modality imbalance in MLLMs. **Right:** Average visual attention score distribution versus model size for different MLLM fine-tuning methods. The plot suggests that methods achieving better balanced intrinsic modality tend to require fewer trainable parameters.

such as image understanding (Sidorov et al., 2020), visual question answering (Goyal et al., 2017a; Lu et al., 2022; Hudson & Manning, 2019), and instruction following (Liu et al., 2023).

In the visual instruction tuning stage, due to its high computational cost, researchers have pursued two primary strategies. One approach focuses on refining data selection methodologies (Liu et al., 2024c; McKinzie et al., 2024) to reduce redundancy and optimize the training dataset, though this process remains expensive and time-consuming. A more common strategy goes to employ Parameter-Efficient Fine-Tuning (PEFT) methods, such as LoRA (Hu et al., 2021), aiming to reduce the number of trainable parameters, thereby making visual instruction tuning more computationally feasible (Liu et al., 2024a; Zhou et al., 2024a). However, even with PEFT methods like LoRA, large-scale MLLMs remain prohibitively expensive to fine-tuning.

This raises a critical question: is there any further possibility to reduce more trainable parameters so that the visual instruction tuning can be further improved? Our research offers a novel viewpoint by focusing on the intrinsic modality imbalance within MLLMs. A closer analysis uncovers an imbalance in output attention computation (Chen et al., 2024a), where textual information tends to dominate the attention distribution during output generation. Specifically, we investigate this issue by analyzing attention score distributions, which evaluates the balance between text and visual modalities. As shown in Figure 1, visual representations are significantly underutilized during visual instruction tuning. More importantly, our analysis reveals that achieving a better balance between these modalities can substantially reduce the number of trainable parameters required for fine-tuning. Hereby we suppose that *intrinsic modality rebalance is the Midas touch to unlock further reductions in the number of trainable parameters.*

To address this challenge, we introduce Modality Linear Representation-Steering (MoReS) to optimize visual instruction tuning, significantly reducing the number of trainable parameters while maintaining equivalent performance. Unlike full fine-tuning, which modifies the entire model, or other popular PEFT methods such as LoRA (Hu et al., 2021), OFT (Qiu et al., 2023), Adapter (Houlsby et al., 2019), and IA3 (Liu et al., 2022), MoReS focuses solely on steering the visual representations. Specifically, our approach freezes the entire LLM during visual instruction tuning to preserve its capabilities in the textual modality. Instead of fine-tuning the full model, we introduce a simple linear transformation to steer visual representations in each layer. This transformation operates within a subspace after downsampling, where visual representations encode rich semantic information in a compressed linear subspace (Zhu et al., 2024; Shimomoto et al., 2022; Yao et al., 2015). By continuously steering visual representations across layers, MoReS effectively controls the output generation process, yielding greater attention inclined to visual modality.

To validate the efficacy of our proposed MoReS method, we integrated it into MLLMs of varying scales (3B, 7B, and 13B parameters) during visual instruction tuning, following the LLaVA 1.5 (Liu et al., 2024a) training recipe. The resulting models, collectively termed LLaVA Steering, achieved competitive performance across three visual benchmarks and six visual question-answering tasks, while requiring 287 to 1,150 times fewer trainable parameters than LoRA, depending on the specific training setup.

In our experiments, we observed the need for a comprehensive framework to systematically analyze and compare various model architectures and training strategies in MLLMs. The wide range of design choices and techniques makes it difficult to standardize and understand the interplay between these components. Evaluating each method across different open-source models is time-consuming and lacks consistency due to implementation differences, requiring extensive data preprocessing and careful alignment between architectures and training recipes. To address this issue, we developed the LLaVA Steering Factory, a flexible framework that reimplements mainstream vision encoders, multi-scale LLMs, and diverse connectors, while offering customizable training configurations across a variety of downstream tasks. This framework simplifies pretraining and visual instruction tuning, minimizing the coding effort. Additionally, we have integrated our attention score distribution analysis into the LLaVA Steering Factory, providing a valuable tool to the research community for further studying intrinsic modality imbalance in MLLMs.

Our work makes the following key contributions to the field of MLLMs:

1. First of all, we propose Modality Linear Representation-Steering (MoReS), a novel method that addresses intrinsic modality imbalance in MLLMs by steering visual representations through linear transformations within the visual subspace, effectively mitigating the issue of text modality dominating visual modality.

2. In addition, we present LLaVA Steering, where with different sizes (3B/7B/13B), three real-world LLaVA MLLMs consisting of different model components are composed by integrating the proposed MoReS method into visual instruction tuning. LLaVA Steering models based on MoReS method achieve comparable performance across three visual benchmarks and six visual question-answering tasks, while requiring 287 to $1,150$ times fewer trainable parameters.

3. Last but not least, we develop the LLaVA Steering Factory, a flexible framework designed to streamline the development and evaluation of MLLMs with minimal coding effort. It offers customizable training configurations across diverse tasks and incorporates tools such as attention score analysis, facilitating systematic comparisons and providing deeper insights into intrinsic modality imbalance.

## 2 RELATED WORK

**Integrating Visual Representation into LLMs:** To leverage pre-trained large language models (LLMs) for understanding visual instructions and generating responses, researchers have introduced cross-attention mechanisms to integrate image information into the language model. Notable examples include models such as LLaMA 3-V (Dubey et al., 2024), IDEFICS (Laurençon et al., 2023), and Flamingo (Awadalla et al., 2023; Alayrac et al., 2022). These models typically follow a two-stage training process: pretraining on large-scale image-text datasets, followed by supervised fine-tuning (SFT) with carefully curated high-quality data. During this process, the self-attention layers in the LLM decoder are kept frozen, with only the cross-attention and perceiver layers updated, ensuring that the text-only performance remains intact.

Another prominent approach employs a decoder-only architecture, as seen in models like the LLaVA family (Liu et al., 2024b;a; 2023), BLIP (Xue et al., 2024; Li et al., 2023a), and Qwen-VL (team, 2024; Bai et al., 2023). These models also follow the pretraining and visual instruction tuning paradigm. In the pretraining stage, a randomly initialized connector is trained while keeping the LLM frozen. However, recent studies (Bai et al., 2023; Chen et al., 2023) have demonstrated scenarios where both the projector and vision encoder are jointly trained during pretraining. Given the limited capacity of adapter modules, it is common to unfreeze the LLM during visual instruction tuning, while keeping the vision encoder frozen.

NVLM (Dai et al., 2024) represents a hybrid approach, combining elements of both the cross-attention and decoder-only architectures. In contrast, vision-encoder-free methods, as explored by models like Fuyu (Bavishi et al., 2023), SOLO (Chen et al., 2024b), and EVE (Diao et al., 2024), directly integrate visual information into LLMs at the pixel level, foregoing traditional vision encoders altogether.

While these approaches have advanced the integration of visual representations into LLMs, they still face significant challenges in the computational demands of visual instruction tuning, motivating further exploration into more efficient methods.

**Visual Instruction Tuning:** Fine tuning of multimodal large language models (MLLMs) for downstream tasks has gained considerable attention, but remains computationally expensive due to large-scale visual instruction datasets and model sizes (Wang et al., 2022). To tackle this challenge, recent advancements have introduced parameter-efficient fine-tuning (PEFT) methods (Houlsby et al., 2019; Li & Liang, 2021), such as LoRA (Hu et al., 2021), enabling more efficient visual instruction tuning.

However, many of these PEFT methods primarily focus on optimizing weights but ignore the intrinsic representation imbalance during visual instruction tuning, thus cannot further reduce the required trainable parameters. This means to look for other novel approaches that can improve the efficiency and effectiveness of visual instruction tuning.

**Representation Steering:** Recent studies (Singh et al., 2024; Avitan et al., 2024; Li et al., 2024; Subramani et al., 2022) have demonstrated that the representations induced by pre-trained language models (LMs) encode rich semantic structures. Steering operations within this representation space have shown to be effective in controlling model behavior. Unlike neuron-based or circuit-based approaches, representation steering manipulates the representations themselves, providing a clearer mechanism for understanding and controlling the behavior of MLLMs and LLMs. For example, (Zou et al., 2023) explores representation engineering to modify neural network behavior, shifting the focus from neuron-level adjustments to transformations within the representation space. Similarly, (Wu et al., 2024a) applies scaling and biasing operations to alter intermediate representations. Furthermore, (Wu et al., 2024b) introduces a family of representation-tuning methods that allows for interpretable interventions within linear subspaces.

In this work, we leverage the concept of representation steering to introduce a novel approach, MoReS, which enhances attention to visual representations, thereby demonstrating superior parameter efficiency compared to baseline PEFT methods (Hu et al., 2021; Houlsby et al., 2019; Liu et al., 2022; Qiu et al., 2023).

## 3 INTRINSIC MODALITY IMBALANCE

This section explores how the two intrinsic modalities—text and vision—are imbalanced during output generation across each layer in MLLMs, as reflected in the attention score distribution. Furthermore, we demonstrate that addressing this modality imbalance effectively during visual instruction tuning can guide the design of methods that require fewer trainable parameters.

We begin with calculating the attention score distribution across both modalities in each layer, as derived from the generated output. In auto-regressive decoding, which underpins decoder-only MLLMs, output tokens are generated sequentially, conditioned on preceding tokens. The probability distribution over the output sequence $\hat{y}$ is formalized as:

$$p(\hat{y}) = \prod_{i=1}^{L} p(\hat{y}_i | \hat{y}_{<i}, R_{\text{text}}, R_{\text{image}}, R_{\text{sys}}) \qquad (1)$$

where $\hat{y}_i$ represents the $i$-th output token, $\hat{y}_{<i}$ denotes the preceding tokens, $R_{\text{text}}$ is the textual representation, $R_{\text{image}}$ is the visual input representation, $R_{\text{sys}}$ accounts for system-level contextual information, and $L$ is the output sequence length.

To quantify modality representation imbalance, we calculate the sum of attention scores allocated to visual representations across all layers in MLLMs. Figure 1 illustrates this imbalance across full

fine-tuning, LoRA, and our proposed MoReS method. The results indicate that textual representations often dominate the output generation process in both full fine-tuning and LoRA.

Further examination of this imbalance across multiple PEFT methods reveals an intriguing trend: methods that make better use of visual representations tend to require fewer trainable parameters during visual instruction tuning.

To validate this observation, we introduce the Layer-wise Modality Attention Ratio (LMAR), formulated as:

$$\text{LMAR}_l = \frac{1}{N} \sum_{i=1}^{N} \frac{\alpha_l^{\text{image},i}}{\alpha_l^{\text{text},i}} \; , \tag{2}$$

where $l$ denotes the layer index, $N$ is the total number of samples, and $\alpha_l^{\text{image},i}$ and $\alpha_l^{\text{text},i}$ are the mean attention scores allocated to visual and textual tokens, respectively, in layer $l$ for the $i$-th sample. LMAR thus provides a robust measure of the attention distribution between modalities, averaged over multiple samples to capture general trends in modality representation across layers.

In our experiments comparing various existing PEFT methods and full fine-tuning, IA3 (Liu et al., 2022) consistently achieves the highest average LMAR score across all layers while requiring the fewest trainable parameters. IA3's superior performance can be attributed to its unique design, which introduces task-specific rescaling vectors that directly modulate key components of the Transformer architecture, such as the keys, values, and feed-forward layers.

Unlike methods that introduce complex adapters or fine-tune all parameters, IA3 optimizes a small but crucial set of parameters responsible for attention and representation learning. By applying element-wise scaling to the attention mechanisms, IA3 effectively re-balances the attention distribution across two intrinsic modalities. This design is particularly beneficial during visual instruction tuning, as it allows the model to dynamically reallocate more attention to visual representations without requiring many trainable parameters.

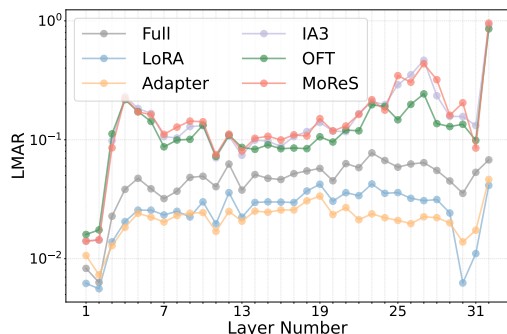

Figure 2: Layer-wise Modality Attention Ratio (LMAR) comparison across training methods, including Full fine-tuning, LoRA, Adapter, IA3, and our MoReS. Our MoReS method (red line) consistently demonstrates the highest LMAR across most layers, with a notable spike in the final layers. Compared with full fine-tuning and mainstream PEFT methods, our MoReS needs the least parameters during visual instruction tuning while achieving superior modality balance.

The identified relationship inspires that if the intrinsic modality imbalance can be addressed, the required number of trainable parameters can be potentially reduced further during visual instruction tuning. This offers a new direction for future improvements in PEFT methods for MLLMs.

## 4    MoReS Method

Based on insights gained from intrinsic modality imbalance, we introduce Modality Linear Representation-Steering (**MoReS**) as a novel method for visual instruction tuning which can rebalance visual and textual representations and achieve comparable performance with fewer trainable parameters.

Our approach is grounded in the linear subspace hypothesis, originally proposed by Bolukbasi et al. (2016), which suggests that information pertaining to a specific concept is encoded within a linear subspace in a model's representation space. This hypothesis has been rigorously validated across numerous domains, including language understanding and interpretability (Lasri et al., 2022; Nanda et al., 2023; Amini et al., 2023; Wu et al., 2024c).

Building upon the intervention mechanisms described in Geiger et al. (2024) and Guerner et al. (2023), we introduce a simple linear transformation that steers visual representations within subspace while keeping the entire LLM frozen during visual instruction tuning. This approach ensures that the language model's existing capabilities are preserved, while continuously guiding the MLLM to better leverage the underutilized visual modality. By steering visual representations across each layer, MoReS effectively rebalances the intrinsic modality and influences the output generation process. Figure 3 provides an illustration of the overall concept and architecture behind MoReS.

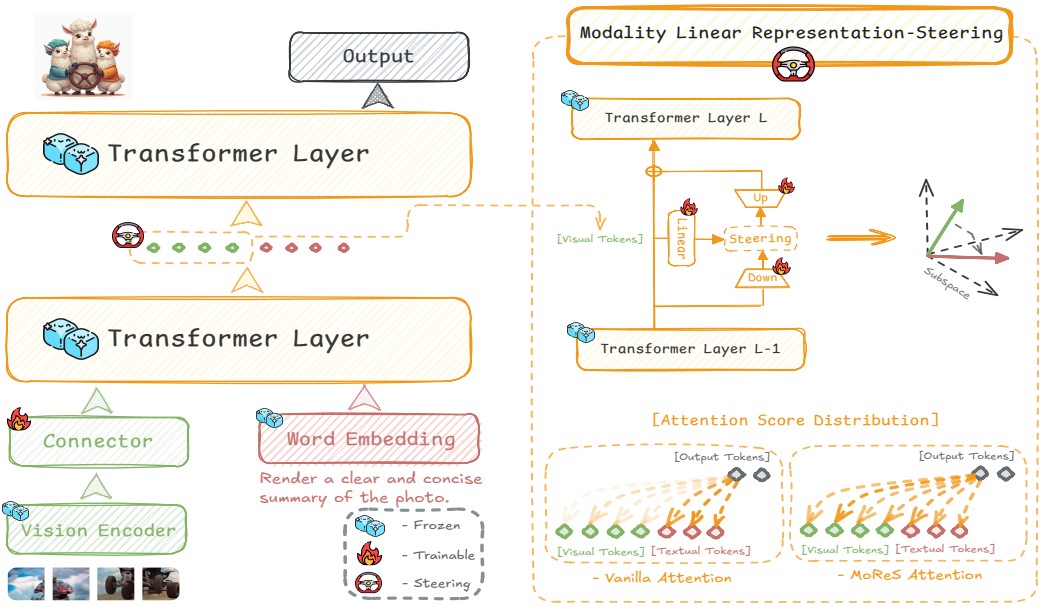

Figure 3: Schematic Overview of Modality Linear Representation-Steering (MoReS): **Left:** The architectural diagram depicts the integration of textual and visual tokens through transformer layers, leading to output token generation. **Right:** The mathematical formulation of MoReS illustrates the steering of visual representations within a subspace, highlighting its impact on output generation. During visual instruction tuning, the parameters of the LLM remain frozen, allowing only the parameters associated with the linear transformation in the steering mechanism to be trainable. With MoReS, the distribution of attention scores becomes more balanced, achieving intrinsic modality balance.

Formally, MoReS method can be formulated as follows: Let $\mathcal{H} = \{h_i\}_{i=1}^N \subset \mathbb{R}^D$ denote the set of visual representations in the original high-dimensional space. We define our steering function MoReS as:

$$\text{MoReS}(h) = W_{\text{up}} \cdot \phi(h) \qquad (3)$$

where $h \in \mathbb{R}^D$ is an input visual representation, $\phi : \mathbb{R}^D \to \mathbb{R}^d$ is a linear transformation function that steers $h$ into a lower-dimensional subspace $\mathbb{R}^d$ ($d < D$), and $W_{\text{up}} \in \mathbb{R}^{D \times d}$ is an upsampling matrix that projects from $\mathbb{R}^d$ back to $\mathbb{R}^D$. The steering function $\phi$ is defined as:

$$\phi(h) = \text{Linear}(h) - W_{\text{down}}h \qquad (4)$$

where $W_{\text{down}} \in \mathbb{R}^{d \times D}$ is a downsampling matrix. To preserve the fidelity of the representation and ensure a bijective mapping between spaces, we impose the following constraint $W_{\text{down}}W_{\text{up}}^T = I_D$. Notably, this steering method can dynamically be applied to specific visual tokens. Further exploration of the impact of different steered token ratios is discussed in Section 5.5.

In Section A.1, we further provide theoretical justification that elucidates how MoReS effectively rebalances the intrinsic modalities while continuously controlling output generation. Additionally, we provide a preliminary estimation of the trainable parameters involved during visual instruction tuning.

In the following sections, we first compose real-world MLLMs (i.e., LLaVA Steering) with three different scales and integrate the proposed MoReS method. Based on the composed real-world

models, we then evaluate how our MoReS method performs within the composed models across several popular and prestigious datasets.

# 5 EXPERIMENTS

We incorporate MoReS into each layer of the LLM during visual instruction tuning, developing LLaVA Steering (3B/7B/13B) based on the training recipe outlined in (Liu et al., 2024a). During visual instruction tuning on the LLaVA-665k dataset, we apply MoReS to a specific ratio of the total visual tokens, specifically using it on only 1% of the tokens.

## 5.1 EXPERIMENT SETTINGS

### 5.1.1 LLAVA STEERING ARCHITECTURES

As illustrated in Figure 3, the architecture of the LLaVA Steering models (3B/7B/13B) consists of three essential components: a vision encoder, a vision connector responsible for projecting visual representations into a shared latent space, and a multi-scale LLM. The three modules are introduced below.

In our experiments, we utilize the Phi-2 2.7B model (Li et al., 2023c) alongside Vicuna v1.5 (7B and 13B) (Zheng et al., 2023b), sourced from our factory, to evaluate the generalizability of our approach across models of varying scales. For vision encoding, we employ CLIP ViT-L/14 336px (Radford et al., 2021) and SigLIP-SO400M-Patch14-384 (Zhai et al., 2023), while a two-layer MLP serves as the connector. Given the inefficiencies of Qformer in training and its tendency to introduce cumulative deficiencies in visual semantics (Yao et al., 2024), it has been largely replaced by more advanced architectures, such as the BLIP series (Xue et al., 2024), Qwen-VL series (team, 2024), and InternVL series (Chen et al., 2024c), which were previously reliant on Qformer.

### 5.1.2 BASELINE TRAINING METHODS

For comparison, four widely adopted PEFT methods (Adapter, LoRA, OFT and IA3) are selected as baselines. These methods establish a comparative framework to assess both the performance and efficiency of our proposed approach. Essentially, our MoReS method replaces these four PEFT methods during visual instruction tuning in LLaVA Steering.

**Adapter:** Building on the framework of efficient fine-tuning (Houlsby et al., 2019), we introduce adapter layers within Transformer blocks. These layers consist of a down-projection matrix $\mathbf{W}_{\text{down}} \in \mathbb{R}^{r \times d}$, a non-linear activation function $\sigma(\cdot)$, and an up-projection matrix $\mathbf{W}_{\text{up}} \in \mathbb{R}^{d \times r}$, where $d$ is the hidden layer dimension and $r$ is the bottleneck dimension. The adapter output is computed as:

$$\text{Adapter}(\mathbf{x}) = \mathbf{W}_{\text{up}}\sigma(\mathbf{W}_{\text{down}}\mathbf{x}) + \mathbf{x}, \tag{5}$$

where the residual connection $(+\mathbf{x})$ preserves the pre-trained model's knowledge. This formulation enables efficient parameter updates during fine-tuning, offering a balance between computational efficiency and adaptation capacity while minimally increasing the model's complexity.

**LoRA:** We employ the low-rank adaptation method (LoRA) proposed by (Hu et al., 2021), which efficiently updates the network's weights with a minimal parameter footprint by leveraging a low-rank decomposition strategy. For a pre-trained weight matrix $W_0 \in \mathbb{R}^{d \times k}$, the weight update is achieved through the addition of a low-rank decomposition, as shown in Equation 6:

$$W_0 + \Delta W = W_0 + BA \tag{6}$$

where $B \in \mathbb{R}^{d \times r}$ and $A \in \mathbb{R}^{r \times k}$ are trainable low-rank matrices, and $r \ll \min(d, k)$.

**OFT:** We utilize the Orthogonal Finetuning (OFT) method, which efficiently fine-tunes pre-trained models by optimizing a constrained orthogonal transformation matrix (Qiu et al., 2023). For a pre-trained weight matrix $W_0 \in \mathbb{R}^{d \times n}$, OFT modifies the forward pass by introducing an orthogonal matrix $R \in \mathbb{R}^{d \times d}$, as illustrated in Equation 7:

$$z = W^{\top}x = (R \cdot W_0)^{\top}x \tag{7}$$

where $R$ is initialized as an identity matrix $I$ to ensure that fine-tuning starts from the pre-trained weights.

**IA3:** Building on the framework established by (Liu et al., 2022), we introduce three vectors $v_k \in \mathbb{R}^{d_k}$, $v_v \in \mathbb{R}^{d_v}$, and $v_{ff} \in \mathbb{R}^{d_{ff}}$ into the attention mechanism. The attention output is computed as:

$$\text{Attention}(Q, K, V) = \text{softmax}\left(\frac{Q(v_k \odot K^T)}{\sqrt{d_k}}\right)(v_v \odot V), \tag{8}$$

where $\odot$ denotes multiplication by element.

## 5.2 MULTI-TASK SUPERVISED FINE-TUNING

To assess the generality of our method, we compare it with the baselines using the LLaVA-665K multitask mixed visual instruction dataset (Liu et al., 2024a). Our evaluation covers multiple benchmarks, including VQAv2 (Goyal et al., 2017b) and GQA (Hudson & Manning, 2019), which test visual perception through open-ended short answers, and VizWiz (Gurari et al., 2018), with 8,000 images designed for zero-shot generalization in visual questions posed by visually impaired individuals. We also use the image subset of ScienceQA (Lu et al., 2022) with multiple-choice questions to assess zero-shot scientific question answering, while TextVQA (Singh et al., 2019) measures performance on text-rich visual questions. MM-Vet (Yu et al., 2023) evaluates the model's ability to engage in visual conversations, with correctness and helpfulness scored by GPT-4. Additionally, POPE (Li et al., 2023b) quantifies hallucination of MLLMs. Finally, we apply the MMMU benchmark (Yue et al., 2024) to assess core multimodal skills, including perception, knowledge, and reasoning.

Following (Zhou et al., 2024b), we define ScienceQA as an unseen task, while VQAv2, GQA, and VizWiz are categorized as seen tasks in LLaVA-665k. To provide a comprehensive evaluation of our MoReS capabilities, we design three configurations: MoReS-Base, MoReS-Large, and MoReS-Huge, each based on different ranks.

We present the results in Table 1, where our MoReS method achieves the highest scores on POPE (88.2) and MMMU (35.8), as well as the second-best performance on ScienceQA (71.9) and MM-Vet (33.3). Notably, MoReS accomplishes these results with 287 to 1150 times fewer trainable parameters compared to LoRA. The scatter plots in Figure 4 further illustrate that MoReS variants (highlighted in red) consistently achieve Pareto-optimal performance, offering an ideal balance between model size and effectiveness.

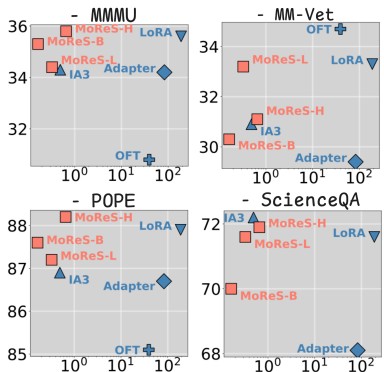

Figure 4: Comparison of parameter count vs. performance for MoReS and other PEFT methods across four benchmarks.

| Model | Method | TP* | VQAv2 | GQA | TextVQA | SciQA-IMG | POPE | MM-Vet | MMMU | Avg |
|-------|--------|-----|-------|-----|---------|-----------|------|--------|------|-----|
| | FT | 2.78B | 79.2 | 61.6 | 57.4 | 71.9 | 87.2 | 35.0 | 38.2 | 61.5 |
| LLaVA Steering-3B | Adapter | 83M | 77.1 | 58.9 | 53.5 | 68.1 | 86.7 | 29.4 | 34.2 | 58.2 |
| | LoRA | 188.74M | 77.6 | 59.7 | 53.8 | 71.6 | 87.9 | 33.3 | 35.6 | 59.9 |
| | OFT | 39.3M | 75.1 | 55.3 | 52.9 | 69.1 | 87.6 | 31.0 | 35.6 | 58.3 |
| | IA3 | 0.492M | 74.5 | 52.1 | 49.3 | 72.2 | 86.9 | 30.9 | 34.3 | 57.1 |
| | MoReS-B | 0.164M | 74.1 | 52.1 | 48.5 | 70.0 | 87.6 | 30.3 | 35.3 | 56.9 |
| | MoReS-L | 0.328M | 74.0 | 51.6 | 49.3 | 71.6 | 87.2 | 33.3 | 34.4 | 57.3 |
| | MoReS-H | 0.655M | 74.2 | 51.8 | 48.3 | 71.9 | 88.2 | 31.1 | 35.8 | 57.4 |

Table 1: Experimental results of Multi-Task Supervised Fine-tuning. For the TP* metric in this evaluation, we focus solely on the trainable parameters within the LLM. While different training strategies are applied to the vision encoder and connector across various recipes, we maintain a consistent training recipe for all models and benchmarks to ensure comparability

## 5.3 Task-Specific Fine-tuning

We evaluate the task-specific fine-tuning capabilities of our MoReS method in comparison to other tuning methods on multiple visual question answering datasets: (1) ScienceQA-Image (Lu et al., 2022), (2) VizWiz (Gurari et al., 2018), and (3) IconQA-txt and IconQA-blank (Lu et al., 2021).

We present the results in Table 2, showing that MoReS achieves 1200 times fewer trainable parameters compared to LoRA and 3 times fewer than the previous best, IA3, while maintaining comparable performance or an acceptable decline of less than 3%. These results show that MoReS can succeed at Task-Specific Fine-tuning, even unseen tasks during its multitask visual instruciton tuning stage.

| Model | Method | TP* | SciQA-IMG | VizWiz | IconQA-txt | IconQA-blank |
|---|---|---|---|---|---|---|
| LLaVA Steering-3B | Adapter | 83M | 92.3 | 62.9 | 93.5 | 95.8 |
| | LoRA | 188.7M | 93.9 | 61.6 | 93.9 | 96.5 |
| | OFT | 39.32M | 86.3 | 42.0 | 87.8 | 42.0 |
| | IA3 | 0.492M | 90.2 | 58.4 | 84.5 | 94.7 |
| | MoReS-B | 0.164M | 89.7 | 59.2 | 84.0 | 94.2 |
| LLaVA Steering-7B | Adapter | 201.3M | 82.7 | 59.7 | 72.1 | 71.6 |
| | LoRA | 319.8M | 87.6 | 60.6 | 77.7 | 70.2 |
| | OFT | 100.7M | 78.3 | 55.1 | 19.4 | 22.7 |
| | IA3 | 0.614M | 83.8 | 54.3 | 65.1 | 70.4 |
| | MoReS-B | 0.262M | 83.6 | 54.2 | 64.2 | 70.2 |
| LLaVA Steering-13B | Adapter | 314.6M | 87.9 | 61.4 | 78.2 | 73.0 |
| | LoRA | 500.7M | 92.1 | 62.0 | 80.2 | 73.2 |
| | OFT | 196.6M | 82.7 | 59.5 | 3.4 | 22.3 |
| | IA3 | 0.963M | 90.5 | 54.6 | 73.8 | 71.7 |
| | MoReS-B | 0.410M | 89.5 | 54.3 | 74.9 | 71.5 |

| Scale | Method | TP* | SciQA-IMG | VizWiz | IconQA |
|---|---|---|---|---|---|
| Small | FT | 2.78B | 33.8 | 51.2 | 68.1 |
| | Adapter | 83M | 81.0 | 57.4 | 72.4 |
| | LoRA | 188.74M | 84.0 | 58.5 | 74.2 |
| | OFT | 39.32M | 79.2 | 43.2 | 35.9 |
| | IA3 | 0.492M | 79.9 | 50.5 | 73.0 |
| | MoReS-L | 0.328M | 78.2 | 55.0 | 69.7 |
| Medium | FT | 2.78B | 78.2 | 58.9 | 92.2 |
| | Adapter | 83M | 92.1 | 60.6 | 93.2 |
| | LoRA | 188.74M | 92.9 | 60.5 | 92.7 |
| | OFT | 39.32M | 86.4 | 44.4 | 45.5 |
| | IA3 | 0.492M | 91.9 | 57.1 | 90.6 |
| | MoReS-L | 0.328M | 92.1 | 56.6 | 89.9 |
| Large | FT | 2.78B | 88.9 | 59.4 | 95.7 |
| | Adapter | 83M | 92.4 | 61.3 | 95.2 |
| | LoRA | 188.74M | 93.9 | 61.8 | 96.0 |
| | OFT | 39.32M | 86.4 | 44.2 | 43.7 |
| | IA3 | 0.492M | 90.3 | 57.9 | 93.8 |
| | MoReS-L | 0.328M | 89.8 | 57.7 | 93.5 |

Table 2: Results of Task-Specific Fine-tuning, where higher values correspond to better performance.

Table 3: Results of multi-scale tasks.

## 5.4 Multi-scale Data Fine-tuning

During visual instruction tuning, the scale of specific task datasets can vary significantly. To gain a comprehensive understanding of our method compared to other training approaches, we follow the methodology of (Chen et al., 2022) and randomly sample 1K, 5K, and 10K data points from each dataset, defining these as small-scale, medium-scale, and large-scale tasks, respectively. Given the limited resources available, we choose MoReS-L for fine-tuning.

Table 3 demonstrates that MoReS exhibits strong capabilities across all scales. Notably, in small-scale tasks, MoReS outperforms full fine-tuning performance while using only 575 times fewer parameters than LoRA and 8,475 fewer than full fine-tuning. In contrast, methods like OFT and IA3 fail to surpass full fine-tuning despite utilizing significantly more parameters. This result underscores the practicality of MoReS in real-world scenarios where data collection can be challenging, suggesting that MoReS is suitable for multi-scale visual instruction tuning.

## 5.5 Ablation Studies

To gain deeper insights into our MoReS method, we conduct ablation studies focusing on its subspace choice and steered visual token ratio. We use LLaVA Steering-3B model as our baseline for comparison. Table 4 summarizes the results of two types of ablations.

First, concerning the choice of subspace rank, we found that a rank of 1 achieves the highest average performance of 81.8 across four visual tasks while also requiring the fewest parameters, specifically 0.164M. Second, regarding the steered visual token ratio, we varied this parameter from 100% (dense steering) to 1% (sparse steering). The results indicate that a ratio of 1% is optimal, yielding the best or near-optimal performance on four benchmarks while also significantly reducing inference overhead due to its sparse steering approach.

# 6 LLaVA Steering Factory

We identified a pressing need for a comprehensive framework to systematically analyze and compare various model architectures and training strategies in MLLMs. The diversity of design choices and

| Subspace Rank | TP* | SciQA-IMG | VizWiz | IconQA-txt | IconQA-blank | Avg | Steered Visual Token Ratio | SciQA-IMG | VizWiz | IconQA-txt | IconQA-blank |
|---|---|---|---|---|---|---|---|---|---|---|---|
| 1 | 0.164M | 89.6 | 59.2 | 84.0 | 94.2 | 81.8 | 1% | 89.7 | 59.2 | 84.0 | 94.1 |
| 2 | 0.328M | 89.7 | 59.2 | 83.9 | 94.0 | 81.7 | 25% | 89.9 | 59.0 | 80.2 | 93.8 |
| 4 | 0.655M | 89.5 | 58.7 | 83.8 | 94.1 | 81.5 | 50% | 88.9 | 59.0 | 79.8 | 92.6 |
| 8 | 1.340M | 89.6 | 58.9 | 83.7 | 93.9 | 81.5 | 100% | 85.8 | 60.5 | 67.7 | 87.8 |

Table 4: Results of the subspace rank choice and steered visual token ratio. The grey shading indicates the best results and our selected parameters.

techniques complicates the standardization and understanding of how these components interact. Evaluating each method across different open-source models is often time-consuming and inconsistent due to implementation differences, which necessitate extensive data preprocessing and careful alignment between architectures and training recipes.

In the LLaVA Steering Factory, we establish standardized training and evaluation pipelines, along with flexible data preprocessing and model configurations. Our framework allows researchers to easily customize their models with various training strategies without the need for additional coding. We implement all mainstream LLMs and vision encoders, including multiple PEFT methods and our proposed MoReS technique. Furthermore, we support a wide range of benchmarks and integrate our intrinsic modality imbalance evaluation. The goal of the LLaVA Steering Factory is to facilitate research in MLLMs, particularly in addressing intrinsic modality imbalance to optimize visual instruction tuning.

An overview of the main components of the LLaVA Steering Factory is provided in Figure 5.

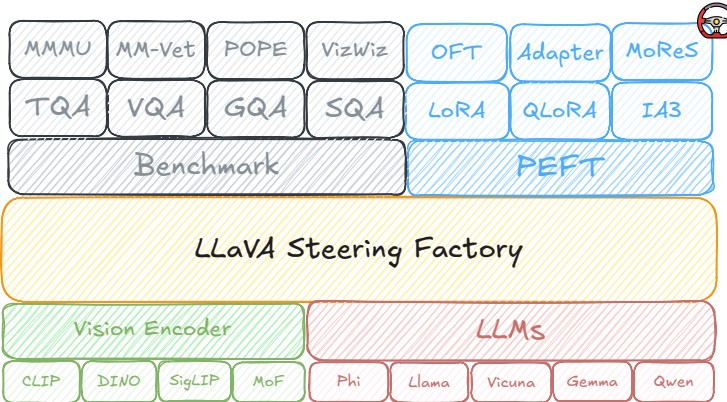

Figure 5: Architectural overview of the proposed LLaVA Steering Factory: A Modular Codebase for MLLMs.

## 7 CONCLUSION

This paper introduces Modality Linear Representation-Steering (**MoReS**), a novel method to significantly reduce the required number of trainable parameters during visual instruction tuning. The key idea behind MoReS is to re-balance visual and textual representations while still maintaining strong performance across a variety of downstream tasks. By integrating MoReS into LLaVA family models, comprehensive evaluation results confirm the effectiveness of the proposed solution. Hence, it further confirms our assertion that intrinsic modality rebalance would represent a promising new approach to optimizing visual instruction tuning.

To facilitate future research in the community, we also present the LLaVA Steering Factory, a versatile framework designed to enhance the development and evaluation of MLLMs with minimal coding effort. This framework enables customizable training configurations for various tasks and integrates analytical tools, such as attention score distribution analysis. This facilitates systematic comparisons among different methods and offers deeper insights into the intrinsic modality imbalance, ultimately contributing to more effective visual instruction tuning.

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

# A  APPENDIX

## A.1  THEORETICAL JUSTIFICATION

Let $x_{\text{text}} \in \mathbb{R}^{d_t}$ be the text input embedding, $x_{\text{image}} \in \mathbb{R}^{d_v}$ be the visual input embedding, $R_{\text{text}} \in \mathbb{R}^D$ be the hidden representation for text, and $R_{\text{image}} \in \mathbb{R}^D$ be the hidden representation for the visual input. Define $W_q, W_k, W_v \in \mathbb{R}^{D \times D}$ as the query, key, and value projection matrices, and $W_o \in \mathbb{R}^{D \times D}$ as the output projection matrix. Let $A \in \mathbb{R}^{N \times N}$ represent the attention matrix, and $y \in \mathbb{R}^V$ be the output logits.

We present a theoretical analysis of the MoReS transformation and its effect on attention redistribution in multimodal models. The hidden representations for text and image inputs are computed as:

$$h_{\text{text}} = f_{\text{text}}(x_{\text{text}}), \quad h_{\text{image}} = f_{\text{image}}(x_{\text{image}}) \tag{9}$$

where $f_{\text{text}}$ and $f_{\text{image}}$ are encoding functions. The attention mechanism is characterized by scores:

$$A_{ij} = \text{softmax}\left(\frac{(h_i W_q)(h_j W_k)^T}{\sqrt{D}}\right) \tag{10}$$

with $W_q, W_k \in \mathbb{R}^{D \times D}$ being query and key projection matrices. Output generation follows:

$$y = W_o(C_{\text{text}} + C_{\text{image}}) \tag{11}$$

where $C_{\text{text}} = \sum_i A_{i,\text{text}}(h_i W_v)$ and $C_{\text{image}} = \sum_i A_{i,\text{image}}(h_i W_v)$.

The core of our approach is the MoReS transformation, defined as:

$$\text{MoReS}(h) = W_{\text{up}} \cdot \phi(h), \quad \text{where} \quad \phi(h) = \text{Linear}(h) - W_{\text{down}} h \tag{12}$$

Here, $W_{\text{up}} \in \mathbb{R}^{D \times d}$, $W_{\text{down}} \in \mathbb{R}^{d \times D}$, and $d < D$. When applied to the image representation, we obtain $h'_{\text{image}} = \text{MoReS}(h_{\text{image}}) + h_{\text{image}}$, leading to updated attention scores:

$$A'_{i,\text{image}} = \text{softmax}\left(\frac{(h_i W_q)(h'_{\text{image}} W_k)^T}{\sqrt{D}}\right) \tag{13}$$

This transformation is key to redistributing attention towards visual inputs. The effect of MoReS on the output can be quantified by examining the change magnitude:

$$\|\Delta y\|_2 = \|W_o(C'_{\text{image}} - C_{\text{image}})\|_2 \leq \|W_o\|_2 \|C'_{\text{image}} - C_{\text{image}}\|_2 \tag{14}$$

where $C'_{\text{image}} = \sum_i A'_{i,\text{image}}(h'_{\text{image}} W_v)$. The significance of this change stems from the MoReS transformation's ability to amplify key visual features. Specifically, $\phi(h)$ extracts salient visual information in a subspace, which is then amplified by $W_{\text{up}}$ in the original space. This process ensures $\|h'_{\text{image}}\|_2 > \|h_{\text{image}}\|_2$, leading to increased $A'_{i,\text{image}}$ values for relevant visual features and larger magnitudes for $(h'_{\text{image}} W_v)$ terms in $C'_{\text{image}}$.

To ensure stability while allowing for this significant attention redistribution, we consider the Lipschitz continuity of the model:

$$\|f(h'_{\text{image}}) - f(h_{\text{image}})\|_2 \leq L \|h'_{\text{image}} - h_{\text{image}}\|_2 \tag{15}$$

where L is the Lipschitz constant. This property bounds the change in the model's output, guaranteeing that the attention redistribution, while substantial, remains controlled and does not destabilize the overall model behavior.

A key advantage of the MoReS approach lies in its parameter efficiency. The transformation introduces $O(Dd)$ parameters, primarily from $W_{\text{up}}$, $W_{\text{down}}$, and the linear transformation in $\phi(h)$. This is significantly less than the $O(D^2)$ parameters required for fine-tuning all attention matrices in traditional approaches. The reduction in trainable parameters not only makes the optimization process more efficient but also mitigates the risk of overfitting, especially in scenarios with limited training data.

In conclusion, our theoretical analysis demonstrates that our MoReS effectively redistributes attention to visual inputs by operating in a carefully chosen subspace. This approach achieves a significant change in output generation while maintaining model stability and requiring fewer parameters than full fine-tuning, offering a balance between effectiveness and efficiency in enhancing visual understanding in MLLMs.

## A.2 IMPLEMENTATION DETAIL

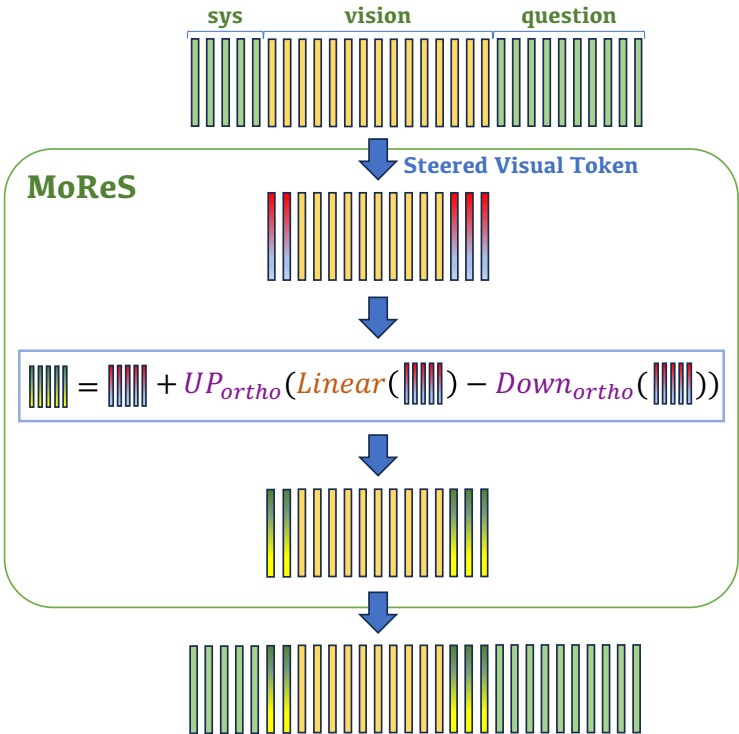

Figure 6: MoReS module flowchart.

Regarding the implementation, we have adopted a highly modular design for the LLM, integrating it with MoReS to enable precise steering at specific token locations. This modular approach ensures that the steering process operates with minimal computational overhead, making it both efficient and scalable. Additionally, the modular nature of this design allows for seamless integration with existing architectures and enables easy customization of steering strategies tailored to specific downstream tasks. To provide further clarity, we include a MoReS module flowchart (Figure 6) and an UML diagram (Figure 7) here, which detail the implementation process.

## A.3 FULL ATTENTION MAPS

In this section, we provide the attention maps (Figure 8) during the decoding process across each layer. Notably, the distribution of visual attention remains sparse in these layers, with only a few tokens carrying the majority of the attention. This sparsity presents an opportunity for token pruning strategies, which can be leveraged to reduce inference overhead and improve computational efficiency. By selectively pruning tokens with lower attention scores, unnecessary computations can be

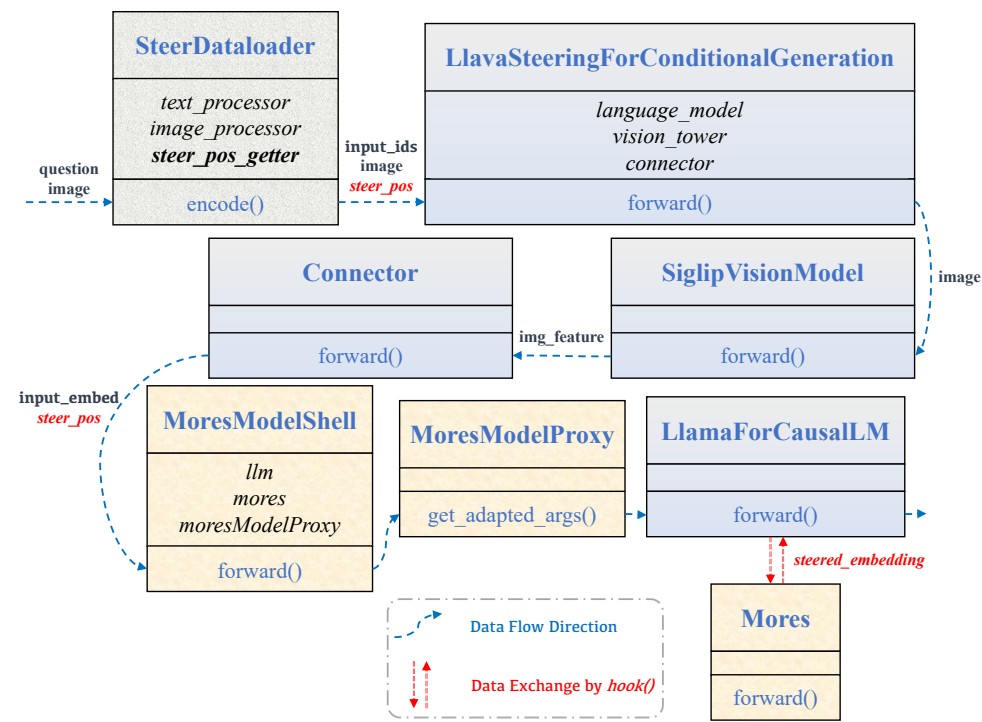

Figure 7: The UML diagram for MoReS

avoided, leading to faster and more efficient inference while maintaining the essential information needed for accurate predictions.

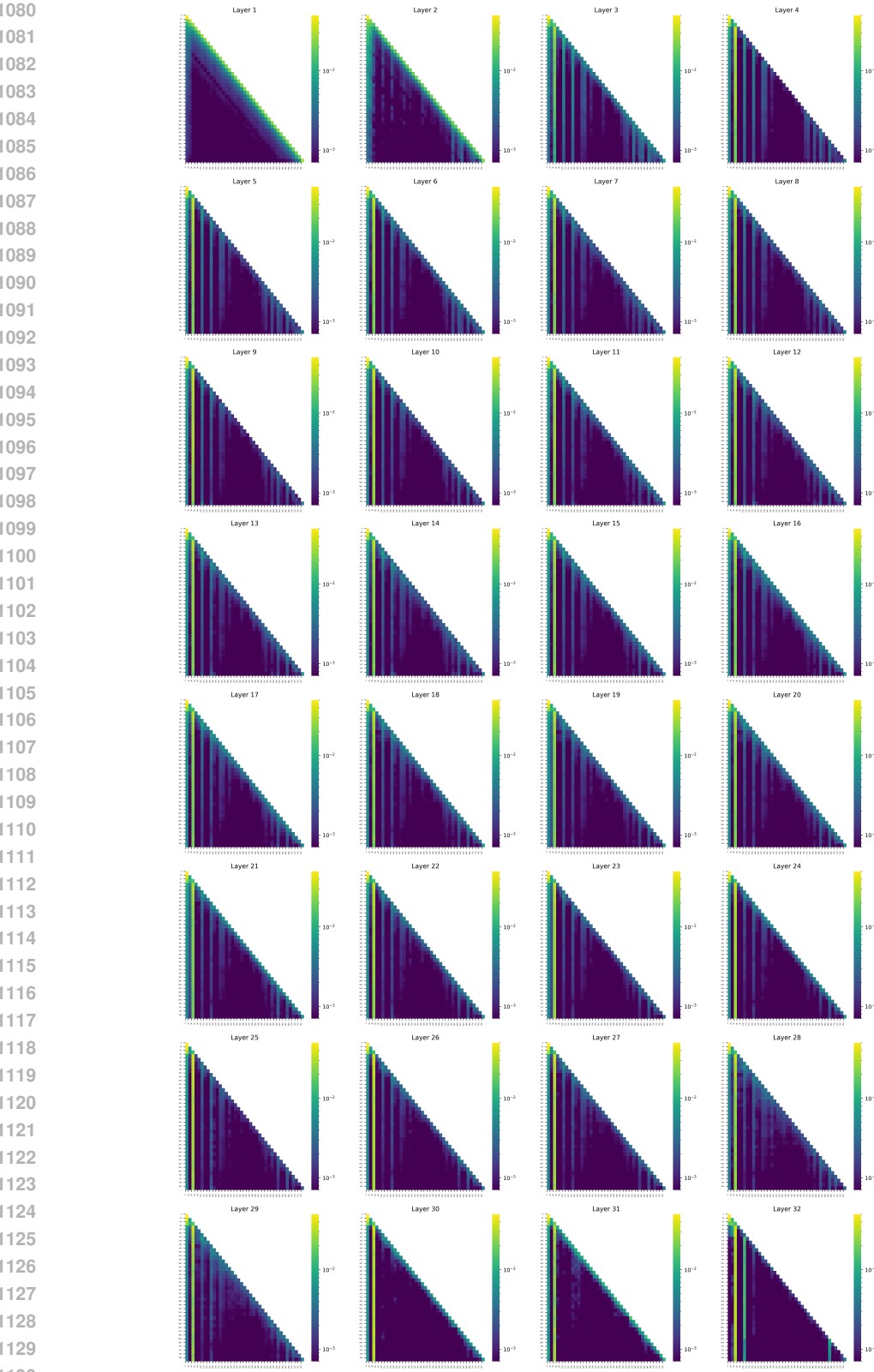

Figure 8: Full Attention Maps of Each Layer

