# OpenReview forum: "Visual Instruction Tuning with 500x Fewer Parameters through Modality Linear Representation-Steering"
_ICLR.cc/2025/Conference — Submitted to ICLR 2025_

### Official Review · Reviewer_NCDD · 2024-10-28

**Soundness:** 2
**Presentation:** 3
**Contribution:** 2
**Rating:** 6
**Confidence:** 3

**Summary:**

This paper presents an approach to visual instruction tuning for MLLMs by introducing Modality Linear Representation-Steering (MoReS). The MoReS method addresses the issue of intrinsic modality imbalance where textual data tends to dominate visual data during tuning by steering visual representations through linear transformations within the model’s layers. This technique reduces the number of trainable parameters required by up to 500 times compared to traditional methods while maintaining competitive performance across visual benchmarks and question-answering tasks. The authors also developed the LLaVA Steering Factory, an open-source framework that enables researchers to customize and experiment with MLLMs more effectively.

**Strengths:**

1. The MoReS provides a solution to balancing visual and textual modalities, reducing computational requirements without sacrificing performance.
2. The MoReS significantly lowers trainable parameters compared to established techniques, like LoRA and full fine-tuning, making it a resource-efficient alternative.
3. This method is tested across multiple model sizes and visual benchmarks, with consistent performance, demonstrating its robustness.
4. The development of the LLaVA Steering Factory offers a tool for the research community.

**Weaknesses:**

1. MoReS aims to improve the visual attention score distribution. However, is a higher visual attention score necessarily better? I hope the authors can conduct an exploratory experiment on this aspect to provide more insight for the readers.
2. Although the model is effective on visual benchmarks, its performance on entirely text-only tasks has not been tested, leaving questions about its generalizability.
3. Table 2 and Table 3 describe the results of different benchmarks. However, there is an anomaly: the performance of the 3B parameter model is significantly higher than that of the 7B and 13B parameter models.
4. FastV[1] finds that attention computation over visual tokens is extremely inefficient in the deep layers of popular LVLMs, suggesting a need for a sparser approach compared to textual data handling. Therefore, this implies that in the deep layers, the role of visual attention is not very significant, raising the question of whether increasing its attention score in these layers is necessary.

[1] An Image is Worth 1/2 Tokens After Layer 2: Plug-and-Play Inference Acceleration for Large Vision-Language Models. ECCV2024.

**Questions:**

See weaknesses.

---

> ### Author Response · Authors · 2024-11-21
> **Response to reviewer NCDD**
>
> We would like to begin by expressing our appreciation to Reviewer NCDD for recognizing the innovative nature of our representation steering approach aimed at tackling modality imbalance. We also value the acknowledgment of our open-source initiative, the LLaVA Steering Factory, which has made a significant contribution to the research community.
>
> ---
>
> **WK1:** Is a higher visual attention score necessarily better?
>
> **A1:** We understand the concerns regarding the necessity of increasing attention scores for performance. We agree that increasing visual attention scores does not always guarantee improved performance across all tasks. However, we believe optimizing visual attention scores strikes a better balance between performance and trainable parameters in MLLMs.
>
> MoReS addresses this by steering representations rather than adjusting parameters. Many existing PEFT and full fine-tuning methods focus on parameter adjustments, often neglecting the importance of visual information. Our analysis shows that this can lead to parameter redundancy and disruption of the model’s inherent knowledge. MoReS retains 100% of the LLM's language capabilities while rebalancing the attention distribution, achieving superior performance compared to IA3, which also increases visual attention but fails to fully preserve the LLM's world knowledge.
>
> By increasing visual attention scores with minimal trainable parameters, MoReS substantially improves the performances on visual understanding tasks such as MMMU, POPE, and SQA.
>
> | Method   | POPE   | SciQA-IMG    |MMMU   |
> |:------|:-----:|:------:|:------:|
> | LoRA | 87.9 | 71.6 | 35.6 |
> | Adapter | 86.7 | 68.1 | 34.2 |
> |  OFT    | 87.6 | 69.1 |   35.6 |
> | IA3 | 86.9 | 72.2 | 34.3 |
> | MoReS | **88.2** | *71.9* | **35.8** |
>
> MoReS achieves state-of-the-art performance on multi-disciplinary tasks requiring advanced reasoning, while reducing hallucination rates and maintaining parameter efficiency. In the revised version, we will provide more detailed comparative results to emphasize the strengths of our approach.

---

> ### Author Response · Authors · 2024-11-21
> **Response to reviewer NCDD**
>
> **WK2:** Text-Only Tasks
>
> **A2:** We sincerely appreciate Reviewer NCDD’s emphasis on evaluating MLLMs' capabilities on text-only tasks. We fully agree that this is a crucial aspect of assessing their generalizability across different modalities.
>
> We quickly tested the text-only tasks given your comment, the results show that MoReS achieves superior performance on two widely recognized LLM benchmarks for text-only tasks. One of the key advantages of MoReS is that it preserves 100% of the world knowledge in LLMs, unlike other tuning methods that may result in performance degradation on text-only tasks [1].
>
> | Text-only Task   | LoRA   | Adapter   |OFT   |IA3     | MoReS  |
> |:------|:-----:|:------:|:------:|:------:|:-----:|
> | HellaSwag |  70.5 | 66.4 |69.1|71.8 |**71.9**|
> | MMLU |  55.3 | 52.9 |54.7| 56.8|**57.0**|
>
> The results clearly demonstrate that MoReS excels in text-only tasks, further emphasizing its ability to retain and effectively leverage the inherent world knowledge stored in LLMs. This capability showcases MoReS’ generalizability not only for multimodal tasks but also for text-dominant tasks. In the revised version of our paper, we will provide a more detailed analysis of these comparative experiments to further emphasize the advantages of MoReS in both text-only and multimodal contexts.
>
> [1] Zhang, Yi-Kai et al. “Wings: Learning Multimodal LLMs without Text-only Forgetting.” NeurIPS 2024.

---

> ### Author Response · Authors · 2024-11-21
> **Response to reviewer NCDD**
>
> **WK3:** 3B Model Outperforms 7B and 13B
>
> **A3:** We appreciate the reviewer’s attention to the performance discrepancies noted in Tables 3. We are quite confident that this is not an anomaly and would like to provide a detailed explanation to clarify this observation:
>
> * **LLM Differences:**
> The 3B parameter model utilizes Phi-2 as its base LLM, which has demonstrated superior performance on several benchmarks due to cleaner training data and higher-quality pretraining compared to Vicuna v1.5, which is used in the 7B and 13B models [1].
>
> * **Visual Encoder Pairing:**
> The 3B model is paired with the SigLIP visual encoder, while the 7B and 13B models use CLIP. This design choice aligns with the methodology of TinyLLaVA, which shows that combining Phi-2 and SigLIP outperforms the 7B and 13B configurations of LLaVA 1.5 across various tasks [2].
>
> * **Fitting Ability:**
> In Table 3, the evaluation focuses on fitting capability on a single dataset. Smaller models like the 3B tend to converge faster and fit better with the same number of training epochs, which aligns with established neural network training dynamics.
>
> This performance gap highlights the importance of our LLaVA Steering Factory framework, which enables researchers to freely explore different combinations of components. This flexibility facilitates deeper insights into how design choices affect performance, making it a valuable tool for future research.
>
> We hope this explanation resolves the reviewer’s concerns and highlights the motivation and flexibility offered by our proposed framework.
>
> [1] Li, Yuan-Fang et al. “Textbooks Are All You Need II: phi-1.5 technical report.” Arxiv
>
> [2] Zhou, Baichuan et al. “TinyLLaVA: A Framework of Small-scale Large Multimodal Models.” Arxiv

---

> ### Author Response · Authors · 2024-11-21
> **Response to reviewer NCDD**
>
> **WK4:** Is Increasing Attention Scores in Deep Layers Necessary?
>
> **A4:** We appreciate the reviewer’s reference to FastV and its observation that the role of visual attention diminishes in deeper layers. We share a similar observation and would like to clarify how increasing attention scores in deep layers is still necessary, without conflicting with the findings of FastV.
> | Steering Layer   | VQAv2   | GQA   |TextVQA   | SciQA-IMG   | POPE   | MM-Vet   |MMMU   | Avg   |
> |:------|:-----:|:------:|:------:|:------:|:-----:|:------:|:------:|:------:|
> | Shallow (0-15) | 74.3   | 51.6 |48.6 | 70.3 | 87.5| 34.9 |34.4 |57.3|
> | Middle (8-23)  | 74.3 | 52.3 | 48.3 | 71.5 | 87.1 | 32.0 |32.6 |56.9 |
> |  Deep (16-31)    | 74.2 | 51.5 |   48.2 | 71.8   | 87.1| 33.3 |   36.7|**57.7**|
>
> * **Experimental Validation on Layer-wise Attention:**
> To further validate our approach, we conducted experiments to assess the performance when applying MoReS to shallow, middle, and deep layers of MLLMs.
> The results show that MoReS applied to deep layers (16-31) consistently achieves the best performance. In MLLMs, the deep layers of the LLM primarily encode high-level semantic features, enabling the model to focus on abstract and complex reasoning tasks. Retaining high attention scores for a few critical tokens in these layers ensures that key visual information remains influential during the inference process. This reinforces the idea that increasing attention scores in the deep layers is both effective and necessary for optimal performance.
>
> * **Visual Attention Distribution and Consistency with FastV:**
> As shown in Figure 1 in our current manuscript, the sum of visual attention scores across layers reveals a significant degradation in deep layers (25-31), which aligns with the findings of FastV. Notably, the distribution of visual attention remains sparse in these layers, with only a few tokens carrying the majority of the attention. This sparsity can further benefit token pruning strategies like FastV. Additional visualization results can be found in the Appendix A.3 for further reference.
>
> We view methods such as FastV as complementary approaches for efficiency optimization, which can work synergistically with MoReS. While MoReS focuses on rebalancing modality representations, FastV’s optimization techniques can be seamlessly integrated with MoReS to enhance computational efficiency, and MoReS can help FastV perform more effective visual token pruning.
> We are actively exploring the incorporation of various efficiency optimization methods, including FastV, into our LLaVA Steering Factory. In the final version of our paper, we will update the relevant content to provide a more detailed breakdown of these experiments and the associated performance improvements.

---

> ### Comment · Reviewer_NCDD · 2024-11-22
> **Response to authors**
>
> Thanks for the rebuttal. After reviewing your response, I have a few questions:
>
> 1. While the improvements reported for MMMU, POPE, and SQA are evident, they appear relatively modest and could potentially be attributed to test-time fluctuations, such as adjusting hyperparameters like the temperature. Could you provide further evidence or analysis to demonstrate the superiority of your approach beyond these minor fluctuations? For instance, are there any scenarios or metrics where your method offers clear advantages?
>
> 2. I noticed that you use different backbones for the 3B, 7B, and 14B models. This could introduce confounding factors and potentially interfere with the robustness and scalability conclusions of your method. Moreover, it may obscure the ability to demonstrate its scalability with respect to scaling laws. Would it be possible to include experiments in the future using the same backbone for all model sizes to ensure a consistent basis for comparison and to better evaluate the scalability of your approach? Additionally, why are the baseline experiments based on the 3B LLM? I believe llava1.5-7B is more general and representative.
>
> 3. If your proposed method is indeed effective, have you considered incorporating it into the finetuning stage? Would this not further improve the model's performance? Exploring this direction could provide additional insights and demonstrate the flexibility and utility of your approach.

---

> > ### Author Response · Authors · 2024-11-26
> > **Response to Reviewer NCDD**
> >
> > Thanks for your prompt feedback! Below, we first summarize each question and then provide our corresponding reply.
> >
> > ---
> >
> > **Q1.1 : Influence of hyperparameter settings to the performance**
> >
> > **A1.1:** Regarding the concern about possible improvements stemming from hyperparameter tuning , we assure you that all hyperparameters were kept consistent across MoReS and other tuning methods , and the evaluation environments were identical to minimize external influences. Hence, the comparisons are fair and the observed improvements are objective results directly attributed to our proposal.
> >
> > **Q1.2: Are there any scenarios or metrics where your method offers clear advantages?**
> >
> > **A1.2:** First of all, we want to clarify that the primary advantage of MoReS lies in its efficiency that achieves up to 1150 times fewer trainable parameters compared to LoRA. As a promising efficient fine-tuning method, our primary goal is not to significantly outperform other PEFT methods on every downstream task. Instead, our primary goal is to provide a highly parameter-efficient alternative. Notably, besides its remarkable parameter efficiency, MoReS also demonstrates superior performance in specific scenarios, thanks to its unique mechanism of modality rebalance and its ability to preserve the world knowledge of the underlying LLM intact.
> >
> > |   | Full |LoRA   | Adapter   |OFT   |IA3     | MoReS  |
> > |:------|:-----:|:-----:|:------:|:------:|:------:|:-----:|
> > | Parameters |  2.78B |188.74M | 83M |39.3M|0.492M|**0.164M**|
> >
> > *Due to character constraints, we will present the two scenarios in the following response.*

---

> > ### Author Response · Authors · 2024-11-26
> > **Response to Reviewer NCDD**
> >
> > *Continued Response to Q1.2*
> >
> > To further illustrate the strengths of our approach, we highlight these two critical scenarios where MoReS significantly enhances MLLM understanding:
> > * **1. Hallucination Mitigation**
> > Hallucination remains a significant challenge in MLLMs. As these models rely heavily on language priors, their strong linguistic bias can overshadow visual information, leading to outputs that favor language coherence over accurate visual grounding. This over-reliance often results in hallucinations—where the model generates outputs inconsistent with the visual context.
> > Our MoReS method outperforms existing tuning methods in mitigating hallucinations, as demonstrated across two popular benchmarks:
> >
> >   - **POPE**: Evaluating object hallucination using the Acc metric.
> >   - **HallucinationBench**: Covering diverse topics and visual modalities with two categories of questions and three main metrics for analysis:
> >     - **Question Categories**:
> >       - Visual Dependent (VD) Questions: Require detailed visual understanding.
> >       - Visual Supplement (VS) Questions: Depend on contextual visual support.
> >     - **Main Evaluation Metrics**:
> >       - Hard Acc: Correctness based on strict adherence to the visual context.
> >       - Figure Acc: Accuracy per figure.
> >       - Question Acc: Overall question accuracy.
> >
> > |   |Metric|Full |LoRA   | Adapter   |OFT   |IA3     | **MoReS**  |
> > |:------|:-----:|:-----:|:-----:|:------:|:------:|:------:|:-----:|
> > | POPE | Acc↑  |87.2| 86.7 |87.9|85.1|86.9|**88.2**|
> > | HallucinationBench |Hard Acc↑ | 37.4 |34.6 |36.2  |33.9|39.3|**42.6**|
> > | HallucinationBench |Figure Acc ↑| 18.5 |16.7 |18.2  |14.1|18.5|**19.4**|
> > |  HallucinationBench|Question Acc↑| 44.4 |43.0 |44.8  |36.2|45.0|**46.1**|
> >
> > As shown in the table, MoReS consistently demonstrates best performance in hallucination mitigation, achieving this with the fewest trainable parameters. This showcases the capability of our proposed method, which effectively rebalances modalities to reduce over-reliance on language priors and generates more robust and trustworthy outputs.
> >
> > * **2. Understanding with Rich World Knowledge**
> > An essential capability of MLLMs is their ability to go beyond surface-level image understanding and leverage joint reasoning with world knowledge to address expert-level questions. To enable this, it is critical to preserve the original world knowledge of the pretrained LLMs without introducing perturbations to their parameters    during fine-tuning.
> > To comprehensively evaluate this capability, our MoReS method was tested on both multimodal and text-only benchmarks:
> >   - **MMMU** (Multimodal Massive Multidiscipline Understanding):
> >     - This benchmark evaluates MLLMs on tasks requiring college-level subject knowledge and deliberate reasoning.
> >     - It emphasizes advanced perception and reasoning using domain-specific knowledge, simulating tasks faced by experts.
> >     - The three most common visual input types in MMMU are diagrams, charts, and tables, which combine rich visual and textual information.
> >   - **MMLU** (Massive Multitask Language Understanding):
> >     - This benchmark covers 57 subjects spanning STEM, humanities, social sciences, and more.
> >     - Tasks range from elementary to advanced professional levels, testing both world knowledge and problem-solving abilities.
> >
> > | Benchmark  | LoRA   | Adapter   |OFT   |IA3     | **MoReS**  |
> > |:------|:-----:|:------:|:------:|:------:|:-----:|
> > | MMMU |  35.6 | 34.2 |35.6|34.3|**35.8**|
> > | MMLU |  55.3 | 52.9 |54.7| 56.8|**57.0**|
> >
> > The results show that MoReS achieves the best performance on both benchmarks, excelling in tasks that require world knowledge and advanced reasoning. This advantage stems from MoReS’ unique ability to preserve 100% of the original world knowledge in pretrained LLMs, unlike other tuning methods that often degrade the LLM’s inherent understanding capabilities.
> > By retaining the world knowledge, MoReS provides a robust solution to enhance the joint reasoning and understanding capabilities of MLLMs, offering a pathway for these models to perform at an expert level in real-world scenarios.
> > We will include this detailed analysis and corresponding results in the revised version of our paper to highlight the strengths of MoReS in enabling MLLMs to tackle complex multimodal and text-only tasks effectively.

---

> > ### Author Response · Authors · 2024-11-26
> > **Response to Reviewer NCDD**
> >
> > **Q2:** Would it be possible to include experiments in the future using the same backbone for all model sizes to ensure a consistent basis for comparison and to better evaluate the scalability of your approach?
> >
> > **A2:** We appreciate your insightful suggestion and fully agree with its importance. To further demonstrate the scalability of our approach, we commit to using the same backbone across all model sizes in our future experiments. We have already started implementing this change and will include the updated results in the final version of the manuscript.
> >
> > In the current setup, we used Phi-2 (3B) and Vicuna v1.5 (7B and 13B) for two main reasons. First, we aimed to decouple our method from a single backbone to avoid potential bias. We are aware that consistent backbones can provide a more precise comparison. Second, the availability of model sizes constrained our choices, as Phi-2 only offers a 3B variant and Vicuna v1.5 is limited to 7B and 13B. This restricted multi-scale comparisons.
> >
> > To resolve this restriction, we will change to adopt unified backbones (e.g., Qwen-2.5 and LLaMA-3.2 series) in our future experiments to comprehensively test scalability. In the revised manuscript, Table 2 will be expanded to include new results from these experiments across different model scales with the same backbone. For the baseline experiments conducted with 7B and 13B models, their results have already been summarized in Table 2 of the current manuscript.
> >
> > **Q3:** If your proposed method is indeed effective, have you considered incorporating it into the finetuning stage?
> >
> > **A3:** First and foremost, we want to clarify that MoReS is indeed a fine-tuning method. In the current manuscript, all experiments are conducted within the supervised fine-tuning paradigm. Specifically, we evaluate our method and other tuning approaches during the multi-task instruction stage (Stage 2) and task-specific fine-tuning stage (Stage 3). Additionally, we assess the fine-tuning capability of MoReS under varying resource constraints. These experiments consistently demonstrate the flexibility and utility of MoReS. We will provide further clarification and additional experimental details in the final version to address this point.

---

> > > ### Comment · Reviewer_NCDD · 2024-11-27
> > >
> > > Thanks for your detailed responses. Most concerns have been addressed. I decided to raise my score to 6.

---

### Official Review · Reviewer_jb5P · 2024-11-02

**Soundness:** 3
**Presentation:** 3
**Contribution:** 2
**Rating:** 5
**Confidence:** 4

**Summary:**

I found this paper to be an innovative contribution to the field of multimodal AI. The researchers tackle a fascinating challenge: the imbalance between text and visual processing in multimodal language models.

They've developed a clever approach called MoReS that dramatically reduces the number of parameters needed for visual instruction tuning - we're talking about using 500 times fewer parameters while maintaining comparable performance.

**Strengths:**

In this paper, the researchers have spotted something that others seemed to have missed - there's an inherent imbalance in how these models handle text versus visual information. Meanwhile, the development of the LLaVA Steering Factory provides a valuable contribution to the research community, offering a standardized framework for implementing and evaluating various MLLM architectures and training strategies.

**Weaknesses:**

Overall, I feel that the contributions of this paper are insufficient. The discovery of the modality imbalance is not original, and the proposed method does not demonstrate particularly outstanding performance compared to existing approaches.

At last, I think the LLaVA Steering Factory is not an academic innovation, it's more like an engineering contributions.

**Questions:**

How does the performance of MoReS change when dealing with images that contain both rich textual and visual information? Is there a potential trade-off between visual and textual understanding when rebalancing the modalities?

---

> ### Author Response · Authors · 2024-11-21
> **Response to reviewer jb5P**
>
> First, we would like to extend our gratitude to Reviewer jb5P for acknowledging the importance of our findings regarding modality imbalance in MLLMs. To the best of our knowledge, we are the first to thoroughly investigate this phenomenon when applying different tuning methods. We also deeply appreciate Reviewer jb5P's kind recognition of the LLaVA Steering Factory as a valuable contribution.
>
> ---
>
> **WK1&Q1:** Trade-off between visual and textual understanding when rebalancing the modalities
>
> **A1:** We understand the reviewer jb5P's concern that if we could still achieve good performance on images with both rich visual and textual information after increasing visual attention score. The answer is yes where MoReS will still perform good on tasks with both rich visual and textual information and the reasons are as follows:
>
> MoReS maintains the model's ability to leverage its LLM-derived 100% world knowledge while incorporating visual context more effectively. This enables richer multimodal reasoning especially for images with both rich visual and textual information. Compared to other PEFT methods, MoReS achieves the highest performance on the MMMU benchmark while utilizing the fewest trainable parameters. MMMU is specifically designed to evaluate three essential skills in MLLMs: perception, knowledge, and reasoning. Notably, the predominant image types in this benchmark—diagrams, charts, and tables—contain a blend of rich visual and textual information. Additionally, we evaluated MoReS on the MMLU benchmark, which primarily focuses on textual information understanding. MoReS also outperformed other methods in this evaluation, leveraging its LLM-derived 100% world knowledge without compromising the pre-trained model's inherent capabilities.
> | Benchmark  | LoRA   | Adapter   |OFT   |IA3     | MoReS  |
> |:------|:-----:|:------:|:------:|:------:|:-----:|
> | MMMU |  35.6 | 34.2 |35.6|34.3|**35.8**|
> | MMLU |  55.3 | 52.9 |54.7| 56.8|**57.0**|
>
> We will update the relevant content in the final version of our paper to provide a more detailed breakdown of these experiments.
>
> Thank you again for your insightful suggestions!

---

> ### Author Response · Authors · 2024-11-21
> **Response to reviewer jb5P**
>
> **Q2:** Considered as an engineering Contribution of LLaVA Steering Factory:
>
> **A2:** We thank Reviewer jb5P for recognizing the value of our LLaVA Steering Factory as a significant contribution to the research community. As one of the key engineering and open-source aspects of our paper, LLaVA Steering Factory provides a highly versatile and comprehensive platform to support multimodal research and experimentation.
>
> During our initial exploration of modality imbalance, we found it particularly challenging to verify results across multi-scale and multi-combination MLLMs on various benchmarks, especially when using all mainstream PEFT methods. This has significantly hindered researchers' academic progress. To better advance the research in this area, we developed the LLaVA Steering Factory, adhering to the software principles of modularity, simplicity, and extensibility. The framework enables all configurations to be implemented in a straightforward and unified scripting manner. The main features of the LLaVA Steering Factory are as follows:
>
> * **Broad Support for Mainstream Models and Benchmarks:**
> LLaVA Steering Factory is designed to integrate most mainstream LLM backbones and visual encoders. It supports various training strategies and evaluation across a wide range of benchmarks, including both visual and text-only tasks, offering researchers unparalleled flexibility.
>
> * **Support for Diverse Tuning Methods:**
> The platform includes implementations of multiple tuning methods, including our advanced MoReS framework. These options cater to varying computational resource constraints, making it accessible to researchers with different hardware setups.
>
> * **Motivation and Community Impact:**
> During our development of various MLLMs and tuning strategies, we often encountered inefficiencies and engineering challenges, especially when dealing with different fine-tuning methods and benchmarking requirements. Inspired by LLaMA Factory [1], a highly influential open-source project with over 34k stars on GitHub that fine-tunes over 100 LLMs and provides tools for fine-tuning and benchmarking, we developed LLaVA Steering Factory to address similar needs in the multimodal domain. Our goal is to further facilitate the research community by streamlining the process of building and evaluating MLLMs.
>
> * **Comparison with Previous Contributions:**
> While earlier efforts, such as Tiny LLaVA Factory [2] and Prismatic VLMs [3], have made valuable contributions, LLaVA Steering Factory significantly extends their capabilities. Below, we provide a detailed comparison:
> | Factory  | Multi-scale LLMs   | Diverse Vision Encoders  | Mainstream PEFTs | Text-only Tasks   | Multimodal Benchmarks  |Computational Optimization  | Multiple Training Strategies   |
> |:------|:-----:|:------:|:------:|:------:|:-----:|:------:|:------:|
> | TinyLLaVA    | ✗  | ✓|✗ | ✗| ✓| ✗ |✓ |
> | Prismatic      | ✓ | ✓| ✗ | ✗| ✓ | ✗|✗ |
> |  LLaVA Steering (Ours)    | ✓✓ | ✓ |   ✓ | ✓   | ✓✓| ✓ |   ✓|
>
>
> * **Commitment to Open Science:**
> We are committed to releasing all code and scripts for LLaVA Steering Factory as soon as the double-blind review period ends, ensuring that our contributions can benefit the community immediately.
>
> [1] Zheng, Yaowei et al. “LlamaFactory: Unified Efficient Fine-Tuning of 100+ Language Models.” ACL 2024
>
> [2] Jia, Junlong et al. “TinyLLaVA Factory: A Modularized Codebase for Small-scale Large Multimodal Models.” ArXiv
>
> [3] Karamcheti, Siddharth et al. “Prismatic VLMs: Investigating the Design Space of Visually-Conditioned Language Models.” ICML 2024

---

> ### Author Response · Authors · 2024-11-27
> **Response to reviewer jb5P**
>
> Thank you for your insightful comments and suggestions. We hope the updated manuscript resolves the points you raised and are glad to discuss any additional questions you might have!

---

### Official Review · Reviewer_dR1a · 2024-11-04

**Soundness:** 3
**Presentation:** 2
**Contribution:** 3
**Rating:** 6
**Confidence:** 4

**Summary:**

The paper introduces Modality Linear Representation-Steering (MoReS), a parameter-efficient fine-tuning method for multimodal LLMs that aims to address modality imbalance between visual and textual representations. The key idea is to steer visual representations through linear transformations similar to LoRA but in a reduced subspace across model layers. They also introduce LLaVA Steering Factory, a framework for developing and evaluating MLLMs.

**Strengths:**

Novel perspective on modality balancing in MLLMs through representation steering.

Comprehensive empirical evaluation across multiple benchmarks and ablations.

Open-source framework (LLaVA Steering Factory) that could benefit the research community

**Weaknesses:**

Unclear correlation between attention scores and performance. Full models performing better despite lower attention scores. The paper lacks causal analysis demonstrating that higher visual attention scores lead to better performance

The comparison to LoRA's parameter count may be misleading since LoRA can be configured with very small ranks. And with adapter methods, most compute for training MMLM still happen in the heavy frozen backbone.

Missing discussion on computational overhead: Unlike LoRA, MoReS transformations appear to require additional runtime compute. Lora can be merged into the backbone matrices but the isolated MoReS seem not.

**Questions:**

Have you explored whether the increased visual attention scores directly contribute to improved performance?

Is there a way to merge/optimize the MoReS transformations into the base model to reduce runtime overhead?

---

> ### Author Response · Authors · 2024-11-21
> **Response to reviewer dR1a**
>
> First, we would like to extend our gratitude to Reviewer dR1a for acknowledging the novelty of our representation steering method and the valuable contribution of LLaVA Steering Factory to the research community. Through comprehensive evaluation conducted via this framework, MoReS has demonstrated exceptional effectiveness.
>
> ---
>
> **WK1&Q1:** Simply increasing visual attention scores may not directly enhance performance across all tasks.
>
> **A1:** Thank you for the reviewer dR1a’s feedback! We understand the concerns regarding the Correlation Between Attention Scores and Performance. We believe that optimizing visual attention scores can help strike a better trade-off between performance and trainable parameters in MLLMs.
>
> MoReS addresses this trade-off by focusing on representation steering rather than parameter tuning. Most existing PEFT methods and full fine-tuning approaches primarily adjust parameters, often overlooking the critical role of visual information. Through our analysis, this oversight can lead to parameter redundancy and unintended disruptions to the LLM’s inherent world knowledge.
>
> By rebalancing visual attention scores with the fewest trainable parameters, we do observe a direct improvement on several visual understanding benchmarks, including MMMU, POPE, and SQA.
>
> | Method   | POPE   | SciQA-IMG    |MMMU   |
> |:------|:-----:|:------:|:------:|
> | LoRA | 87.9 | 71.6 | 35.6 |
> | Adapter | 86.7 | 68.1 | 34.2 |
> |  OFT    | 87.6 | 69.1 |   35.6 |
> | IA3 | 86.9 | 72.2 | 34.3 |
> | MoReS | **88.2** | *71.9* | **35.8** |
>
> Through the enhanced visual attention scores, MoReS achieves state-of-the-art performance on diverse, multi-disciplinary tasks that require college-level subject knowledge and deliberate reasoning. Furthermore, it significantly reduces hallucination rates while maintaining parameter efficiency. In the revised version, we will provide a more detailed description of these comparative experiments to further highlight the advantages of our approach.

---

> ### Author Response · Authors · 2024-11-21
> **Response to reviewer dR1a**
>
> **WK2:** LoRA Configuration Setting
>
> **A2:** We understand that LoRA can be configured with a smaller rank to reduce trainable parameters. However, to ensure a fair and robust comparison, we selected a rank of 128 for LoRA in our work based on the following considerations:
> | LoRA Rank|  TP | VQAv2   | GQA   |TextVQA   | SciQA-IMG   | POPE   | MM-Vet   |MMMU   | Avg   |
> |:------:|:------:|:-----:|:------:|:------:|:------:|:-----:|:------:|:------:|:------:|
> | 4  | 5.90M  | 77.3  | 58.7 |53.1 | 71.7 | 89.3| 30.5 |38.0 |59.7|
> | 8   |  11.80M | 77.4 | 59.3 | 53.3 | 71.2 | 87.1 | 32.5 |37.1 |59.7 |
> |  64  |94.37M  | 77.1 | 59.8 |   53.7 | 72.3   | 87.8| 31.0 |   37.2|59.8|
> |  128 | 188.74M  | 77.6  | 59.7 | 53.8| 71.6   | 87.9 | 33.3 |   35.6 | **59.9** |
>
>
> * **Empirical Evaluation of LoRA Rank Settings:**
> We conducted a series of experiments to identify the optimal LoRA rank configuration for our comparisons.
> The results of these ablation studies indicate that a rank of 128 consistently delivers the best performance, establishing a strong baseline. Even with smaller ranks, such as 4, LoRA still requires significantly more trainable parameters (5.90M) than MoReS (0.16M), highlighting the parameter efficiency of our approach.
>
> * **Alignment with Influential Prior Work:**
> Several influential works in this domain, such as LLaVA 1.5 [1] and TinyLLaVA [2], also adopt a LoRA rank of 128. Using the same configuration ensures consistency and comparability with widely accepted benchmarks and methodologies in the field.
>
> Based on these two considerations, we selected a LoRA rank of 128 for our experiments. In the revised version of our paper, we will include these supplementary results and further clarifications to provide a more comprehensive justification for this choice.
>
> [1] Liu, Haotian et al. “Improved Baselines with Visual Instruction Tuning.” CVPR 2024
>
> [2] Zhou, Baichuan et al. “TinyLLaVA: A Framework of Small-scale Large Multimodal Models.” Arxiv

---

> ### Author Response · Authors · 2024-11-21
> **Response to reviewer dR1a**
>
> **WK3:** Potentially High Runtime Overhead
>
> **A3:** Thank you for your valuable feedback! We understand that inference overhead is an important consideration when evaluating the efficiency of MLLMs. We are aware of the overhead incurred in MoReS, however, there are several measures to decrease the overheads effectively.
>
> Unlike LoRA, where the learned weights can be merged into the model’s original parameters to achieve zero computational overhead during inference, MoReS requires the linear transformation layers to remain in the computation graph of the MLLM. While this introduces a small overhead, we have worked to minimize it effectively.
>
> To mitigate runtime overhead, we performed several experiments focusing on key factors: Subspace Rank, Steered Visual Token Rate and Steering Layer Configuration.
> These experiments helped us reduce the additional computational burden. Specifically, by choosing a 1% Steered Visual Token Rate, a Subspace Rank of 1, and employing a sparse Steering Layer Configuration, we achieved the minimum runtime overhead of about 0.08 seconds each sample. This is significantly lower compared to other PEFT methods, such as Adapter (0.3 seconds) and OFT (2.8 seconds).
>
> We will update the relevant content in the final version of our paper to provide a more detailed breakdown of these experiments and the resulting performance improvements.
> Thank you again for your insightful suggestions!

---

> ### Author Response · Authors · 2024-11-27
> **Response to reviewer dR1a**
>
> Thank you for your detailed review and valuable feedback. We hope our revisions have sufficiently addressed your concerns, and we’re happy to provide further clarifications if needed!

---

> > ### Comment · Reviewer_dR1a · 2024-12-03
> >
> > I thank authors for the detailed rebuttal. It is worth pursuing PEFT algorithms from more vision-centric perspective for V-L models. The proposed method is noteworthy as it requires much less memory than LoRA. However, from the rebuttal so far, I am still not convinced that the proposed method is better because
> >
> > 1) the rebuttal still does not answer the question on how does rebalancing visual attention scores contribute to performance. The table does not list the visual attention score of alternative models. It is very interesting to see the stats of the visual attention in the paper, but there is no evidence so far that we *need* to rebalance the score.
> >
> > Herefore, I am maintaining my score.

---

> > > ### Author Response · Authors · 2024-12-03
> > > **Response to reviewer dR1a**
> > >
> > > Thank you, Reviewer dR1a, for acknowledging the efficiency of our proposed MoReS.
> > >
> > > It is a pleasure that some of your earlier concerns could be resolved. Unfortunately, some other concerns were not addressed but we acknowledge and will respect your decision. If allowed, let us try again to clarify your remaining concerns.
> > >
> > > The primary advantage of MoReS lies in its exceptional efficiency, achieving up to 1150 times fewer trainable parameters compared to LoRA by rebalancing modality scores. Therefore, our primary objective is not to significantly outperform other PEFT methods on every downstream task but rather to offer a highly efficient alternative.
> > >
> > > Moreover, beyond that, MoReS also demonstrates superior performance in two key scenarios: **Hallucination Mitigation** and **Understanding with Rich World Knowledge**. This is enabled by its unique modality rebalance mechanism and its ability to preserve the underlying LLM’s world knowledge intact (as detailed in our response to Reviewer NCDD).
> > >
> > > Though we were not aiming to change your mind, we hope the brief clarification can at least resolve your remaining concerns. Thank you once again for your constructive feedback.

---

### Official Review · Reviewer_vteu · 2024-11-05

**Soundness:** 3
**Presentation:** 3
**Contribution:** 3
**Rating:** 8
**Confidence:** 5

**Summary:**

This paper introduces Modality Linear Representation-Steering (MoReS), a new parameter-efficient fine-tuning method for Multimodal Large Language Models (MLLMs) that addresses the imbalance between text and visual modalities during visual instruction tuning. The authors note that existing methods like LoRA underutilize visual information, resulting in text-dominant outputs. MoReS remedies this by steering visual representations through linear transformations in a lower-dimensional subspace at each layer, re-balancing modality influence without altering core LLM parameters. Experiments with LLaVA Steering models show that MoReS matches LoRA's performance on various visual benchmarks and VQA tasks while using up to 500 times fewer trainable parameters. Additionally, the paper introduces the LLaVA Steering Factory, a flexible framework for developing and evaluating MLLMs, which aids in research on modality imbalance and simplifies the integration of different model components.

**Strengths:**

1. The paper is well-written and easy to understand. The motivation for addressing modality imbalance is clearly stated, and the MoReS method is illustrated with diagrams and mathematical formulations. The experimental setup and results are also clearly presented.

2. The experimental evaluation covers multiple visual benchmarks and VQA tasks, comparing LLaVA Steering with established baselines and demonstrating its effectiveness.

3. LLaVA Steering models demonstrate a substantial reduction in trainable parameters (up to 500x less than LoRA) while maintaining comparable performance, making visual instruction tuning more accessible.

4. I like the idea of designing "steering" mechanism to balance the multi-modal clues. The use of linear transformations within a subspace to steer visual representations is a clever and seems an efficient way to achieve modality balance without significantly increasing the number of trainable parameters.

**Weaknesses:**

1. The motivation presented is unconvincing. In my view, relying more on visual information may not yield correct answers for certain Visual Question Answering (VQA) tasks, as many of these tasks require the MLLM to possess relevant world knowledge in addition to visual cues. It would be helpful to provide specific examples that illustrate how the "steering" module functions across different tasks that require either visual or textual information. In the current version, while experimental results are provided, they are insufficient on their own; understanding the underlying reasons— the "why"—is equally important for a persuasive argument.

2. Insufficient ablation studies. While the comparison with LoRA provides a good baseline, the paper would benefit from more ablation studies to analyze the contribution of different components of MoReS. For example, exploring the impact of removing the linear transformations in certain layers or varying the downsampling rate could provide valuable insights.

3. Although I like the general idea of "steering" in terms of different modalities, the downsampling function $\phi$ within MoReS lacks a detailed explanation of its implementation and the rationale behind the chosen dimensionality. This makes it difficult to fully understand the mechanism and reproduce the results. Further clarification on the impact of subspace dimensionality on performance would strengthen the paper. Furthermore, as with the reparameterization trick underlying LoRA, this work should also include a theoretical analysis of the proposed method.

**Questions:**

see "weaknesses"

---

> ### Author Response · Authors · 2024-11-21
> **Response to reviewer vteu**
>
> First, we would like to extend our gratitude to reviewer vetu for acknowledging the efficacy of our MoReS method, a subspace-based representation steering approach.
> Below, we first summarize the key concerns of every weakness point and then provide our reply correspondingly.
>
> ---
> **WK1:** The "steering" module might face challenges in achieving consistent performance across tasks that demand varying levels of visual or textual information.
>
> **A1:** Reviewer vetu concerns that relying more on visual information may not perform well for certain tasks which require both world knowledge and visual information. We share Reviewer vetu's perspective in considering the roles that both modalities play in MLLM multitask scenarios and we also considered this when we design MoReS.
>
> Specifically, MoReS preserves 100% of the pre-trained world knowledge in the LLM by neither modifying its parameters nor interfering with textual token inference. This design allows MoReS to excel in understanding both visual and textual information. Unlike many existing methods, which often alter model weights and risk degrading pre-trained knowledge [1], MoReS employs a representation-steering approach to selectively enhance the performance of the visual modality.
>
> To demonstrate that MoReS performs consistently well across tasks with varying modality dependencies, we selected three representative benchmarks:
> - POPE: Visual-dependent.
> - HellaSwag: Textual-dependent.
> - MMMU: Dependent on both modalities.
>
> | Method   | POPE   | HellaSwag   |MMMU   |
> |:------|:-----:|:------:|:------:|
> | LoRA | 87.9 | 70.5 | 35.6 |
> | Adapter | 86.7 | 66.4 | 34.2 |
> |  OFT    | 87.6 | 69.1 |   35.6 |
> | IA3 | 86.9 | 71.8 | 34.3 |
> | MoReS | **88.2** | **71.9** | **35.8** |
>
> The results demonstrate that MoReS performs effectively across tasks requiring varying levels of visual or textual information. In the revised version, we plan to include the new comparison results to clarify MoReS's capability to handle tasks with varying modality dependencies.
>
> [1] Zhang, Yi-Kai et al. “Wings: Learning Multimodal LLMs without Text-only Forgetting.” NeurIPS 2024.

---

> ### Author Response · Authors · 2024-11-21
> **Response to reviewer vteu**
>
> **WK2**: Insufficient ablation studies on the impact of removing linear transformations or varying the downsampling rate
>
> **A2**: Thank you for highlighting the importance of additional ablation studies. We agree that these experiments will strengthen the validation of our method.
>
> In the current manuscript, in fact, we had provided an ablation study on the downsampling rate (Subspace Rank) with Table 5. In addition to that, we now further study this problem with additional experiments to investigate the effect of removing linear transformations in specific layers.
>
> In Table 5 of the current submission, we conducted experiments using different subspace ranks to modify the downsampling rate. The results indicate that MoReS exhibits relatively robust performance across various subspace ranks. Additionally, by selecting a smaller subspace rank, we can further improve computational efficiency.
>
> | Steering Layer   | VQAv2   | GQA   |TextVQA   | SciQA-IMG   | POPE   | MM-Vet   |MMMU   | Avg   |
> |:------|:-----:|:------:|:------:|:------:|:-----:|:------:|:------:|:------:|
> | [0,2,4,...] | 74.1   | 52.0 |48.3 | 71.6 | 87.1 | 32.8 |35.3 |**57.3**|
> | [0,3,6,...]  | 74.1 | 51.7 | 48.1 | 70.7 | 87.0 | 32.7 |33.2 |56.8 |
> |  [0,4,8,...]     | 74.1 | 51.9 |   48.5 | 71.2   | 87.2| 31.5 |   34.4|57.0|
> |  All Layer    | 74.0  | 51.6 | 49.3 | 71.6   | 87.2 | 33.3 |   34.4 | **57.3** |
>
>
> | Steering Layer   | VQAv2   | GQA   |TextVQA   | SciQA-IMG   | POPE   | MM-Vet   |MMMU   | Avg   |
> |:------|:-----:|:------:|:------:|:------:|:-----:|:------:|:------:|:------:|
> | Shallow (0-15) | 74.3   | 51.6 |48.6 | 70.3 | 87.5| 34.9 |34.4 |57.3|
> | Middle (8-23)  | 74.3 | 52.3 | 48.3 | 71.5 | 87.1 | 32.0 |32.6 |56.9 |
> |  Deep (16-31)    | 74.2 | 51.5 |   48.2 | 71.8   | 87.1| 33.3 |   36.7|**57.7**|
> |  All Layer    | 74.0  | 51.6 | 49.3 | 71.6   | 87.2 | 33.3 |   34.4 | 57.3 |
>
>
>
> Furthermore, as shown in the tables here, we conducted experiments applying MoReS with different fixed intervals and also evaluated its performance when applied exclusively to the shallow, middle, and deep layers. These experiments highlight that the choice of steering layers can effectively balance computational efficiency and performance. We suggest that, when using MoReS, it is optimal to apply it to all layers initially to achieve the best performance. Then, by skipping fixed intervals, we can further reduce inference overhead while maintaining performance. Regarding the choice of shallow, middle, and deep layers, we found that applying MoReS to the deep layers yields better performance. We believe that deep layers encode more abstract concepts and are more suitable for steering in the subspace. We will provide these supplementary results to further clarify this aspect in the final version of our paper.

---

> ### Author Response · Authors · 2024-11-21
> **Response to reviewer vteu**
>
> **WK3**: Insufficient explanation of the mechanism behind MoReS
>
> **A3:** Thank reviewer vetu for acknowledging the effectiveness of our steering approach. We understand your concerns regarding the mechanism behind MoReS, particularly the rationale behind the chosen dimensionality, as this could provide stronger support for understanding how MoReS functions.
>
> MoReS is a sparse, token-level representation tuning method, steering fewer than 1% of visual tokens while preserving textual token inference and LLM parameters. For clarity, we have included an UML diagram and a MoReS module flowchart in the Appendix A.2 to detail the implementation. Additionally, we will release the code alongside the LLaVA Steering Factory, ensuring transparency and enabling reproducibility of our method.
>
> Regarding the rationale behind the chosen dimensionality, we had provided both theoretical and empirical analysis in the current manuscript to demonstrate the effectiveness of our chosen dimension.
> * **Theoretical perspective:**
> We posit that MLLMs represent concepts as subspaces in high-dimensional vector spaces, consistent with previous work [2][3][4]. MoReS capitalizes on this by employing a downsampling function that projects selected high-dimensional visual representations into a lower-dimensional subspace using a low-rank projection matrix with orthonormal rows. This approach ensures that the subspace retains critical multimodal task features. Further theoretical details can be found in Appendix A.1.
>
> * **Empirical perspective:**
> We conducted an ablation study on the chosen dimensionality, as presented in Table 5, to empirically validate the rationality of our choice. The evaluation results can confirm the efficacy of the selected dimensionality in maintaining performance while ensuring efficiency.
>
> Thank you again for your valuable feedback! We will update the relevant content accordingly. If you have any further questions/concerns/suggestions, please do let us know.
>
> [2] Maiorca, Valentino et al. “Latent Space Translation via Semantic Alignment.” NeurIPS 2023.
>
> [3] Park, Kiho et al. “The Linear Representation Hypothesis and the Geometry of Large Language Models.” ICML 2024.
>
> [4] Mikolov, Tomas et al. “Linguistic Regularities in Continuous Space Word Representations.” North American Chapter of the Association for Computational Linguistics (2013).

---

> ### Author Response · Authors · 2024-11-27
> **Response to reviewer vteu**
>
> Thank you for your time and thoughtful review. We hope the revisions have addressed your concerns. If there are any remaining questions or updates, we’d be happy to clarify further!

---

> > ### Comment · Reviewer_vteu · 2024-11-28
> >
> > Thank authors for the detailed rebuttal. I believe they made a good rebuttal, where most of my concerns have been solved properly. Therefore, I would like to raise the score to 8.

---

### Meta-Review · Area_Chair_Fvvw · 2024-12-20

**Metareview:**

This work introduces Modality Linear RepresentationSteering (MoReS) to reduce the trainable parameters by balancing vision and language modalities in MLLMs. Moreover, a platform, i.e., LLaVA Steering Factory, is proposed to evaluate
intrinsic modality imbalance for different models. The work received mixed reviews: 8, 6, 5, 6. The major concerns are about the relationship between rebalancing visual attention scores and performance and modest performance compared to baselines. AC read the paper, comments of reviewers and the rebuttal carefully. Compared with the baseline IA3, AC agrees that the proposed method cannot beat its performance significantly with a similar number of learnable parameters according to Tables 1-3. Since the main contribution of this work is for performance, the work cannot be recommended for acceptance with the current empirical results.

**Additional Comments On Reviewer Discussion:**

After rebuttal, Reviewer vteu and NCDD increased the ratings to 8 and 6 respectively since most of their concerns were addressed. However, Reviewer dR1a was not convinced that the proposed method was better and kept the original score.

---

### Decision · Program_Chairs · 2025-01-22

Reject